# Impact of vaccination and risk factors on COVID-19 mortality amid delta wave in Libya: A single center cohort study

Inas Alhudiri[1]☯*, Zakarya Abusrewil[2,3]☯, Omran Dakhil[4], Mosab Ali Zwaik[4], Mohammed Ammar Awn[4], Mwada Jallul[2], Aimen Ibrahim Ahmed[3], Rasha Abugrara[3], Adam Elzagheid[1]

1 Libyan Biotechnology Research Center, Tripoli, Libya, 2 Department of Forensic and toxicology, Faculty of Medicine, University of Tripoli, Tripoli, Libya, 3 National Migration Health, International Organization for Migration, Tripoli, Libya, 4 Souq Thullatha Isolation Center, Tripoli, Libya

☯ These authors contributed equally to this work.
* einasse@gmail.com

## Abstract

### Introduction

The Delta variant has led to a surge in COVID-19 cases in Libya, making it crucial to investigate the impact of vaccination on mortality rates among hospitalized patients and the critically ill. This study aimed to explore the risk factors for COVID-19 mortality and the mortality rates among unvaccinated and vaccinated adults during the Delta wave who were admitted to a single COVID-19 care center in Tripoli, Libya.

### Methods

The study involved two independent cohorts (n = 341). One cohort was collected retrospectively from May 2021-August 2021 and the second cohort was prospectively collected from August 2021-October 2021. Most of the patients in the study became ill during the Delta wave. The two cohorts were merged and analysed as one group.

### Results

Most patients were male (60.5%) and 53.3% were >60 years old. The vast majority of patients did not have a previous COVID-19 infection (98.9%) and were unvaccinated (90.3%). Among vaccinated patients, 30 had received one dose of vaccine and only 3 had received two doses. Among patients who received one dose, 58.1% (18/31) died and 41.9% (13/31) survived. Most patients (72.2%) had a pre-existing medical condition. A multivariable prediction model showed that age >60 years was significantly associated with death (odds ratio = 2.328, CI 1.5–3.7, p-value = <0.0001).

### Conclusion

Our results indicate that previous infection or full vaccination against COVID-19 significantly reduces hospitalization and death. However, a single vaccine dose may not be adequate,

**Data Availability Statement:** The data are submitted with manuscript as Supporting information.

**Funding:** The author(s) received no specific funding for this work.

**Competing interests:** The authors have declared that no competing interests exist.

especially for older individuals and those with underlying medical conditions. High-risk older patients with comorbidities should be fully vaccinated and offered up to date bivalent COVID-19 booster doses.

## Introduction

The novel coronavirus (SARS-CoV-2) emerged in December 2019 and then spread around the globe. In Libya, the first case was identified on March 24th, 2020 by the National Center for Diseases Control (NCDC). The first case was diagnosed in Tripoli in a pilgrim who came from Saudi Arabia; thereafter cases continued to rise [1, 2].

According to reports from the NCDC, up until the time of this manuscript, the total number of confirmed COVID-19 cases was 502,076. The number of people who had received one dose of vaccine was 2,260,427 with 1,177,469 having received 2 doses, and 117,799 with the booster dose [2].

COVID-19 vaccinations have been approved for emergency use worldwide [3]. The humoral immune responses elicited by four vaccines used in Libya were previously evaluated in the general population [4]. This study showed that Sputnik V and AstraZeneca vaccines developed a robust antibody response especially in previously infected individuals. The response was more pronounced than the Sinovac and Sinopharm vaccines.

Vaccination with either one dose of Pfizer BioNtech or AstraZeneca (Vaxzevria) was associated with a substantial reduction in symptomatic infection in adults older than 70 years and with additional protection against severe disease and hospitalization [5]. Patients who had received one dose of AstraZeneca had an extra reduction of hospital admission with severe disease (4). Furthermore, vaccination was associated with lower in-hospital mortality [6].

The onset of this pandemic has overextended the healthcare systems of many countries, particularly the need for intensive care units (ICU) [7]. Therefore, understanding the risk factors and immunopathology of COVID-19 would help clinicians prioritize patients who may become critically ill and prevent the progression of the disease. This would relieve part of the burden on the health care systems and improve mortality since increased admission to the ICU is associated with an increase in mortality. The infection fatality rate (IFR) and risk of death could be variable for each country due to variations in healthcare systems, such as infrastructure, bed capacity, medical staff and availability of medications [8].

In a study of COVID-19 patients in the United States, the factors associated with hospital admission were elderly age, male gender, ethnicity, fever, dyspnea, comorbid diseases, cancer, being a healthcare worker, smoking, and pregnancy [9]. The signs and symptoms associated with the need for ICU care were dyspnea, tachypnea and hypoxia [10].

Prior studies have reported that approximately 14% to 29% of hospitalized COVID-19 patients required intensive care due to acute respiratory distress syndrome (ARDS). The mortality rate ranged from 8.7% to 21% among those patients with pneumonia [11–14].

The case fatality rate, defined as the ratio between confirmed deaths and confirmed cases, was between 1.7%-1.4% in Libya during the study period (May-October 2021) [15]. However, there is limited published data available on the clinical characteristics, risk factors for COVID-19, vaccination status, and impact on mortality among hospitalized patients and the critically ill in Libya. Therefore, the objective of this study was to determine the impact of vaccination and risk factors on mortality rates among hospitalized patients during the Delta wave of COVID-19 in Libya.

## Methods

### Study cohort and design

The study was conducted at Souq Thullatha Isolation Center, a recently established COVID-19 dedicated healthcare isolation center in Tripoli, Libya. Souq Thullatha isolation center has 20 ICU beds, 27 inpatient beds and 18 patient under investigation (PUI) units. All adult patients admitted during the specified period were included. Our study involved two independent cohorts. One cohort was collected retrospectively from May 2021-August 2021 and the second cohort was prospectively collected from August 2021-October 2021. This combination allowed us to obtain a larger number of cases and overcome potential limitations associated with studying a single cohort and increase the generalizability of our findings. The data from the two cohorts were independent, they were exposed to the same factors, they occurred in the same year and most of them were infected during the Delta wave. Hence the two cohorts were merged and analyzed as one group. Therefore, this is a combined retrospective-prospective cohort study where all individuals were admitted to the Souq Thullatha isolation center over a 6 month period from May 2021-October 2021.

Signed written informed consent was obtained from patients, legal representative or next of kin on admission for the prospective cohort. For the retrospective cohort, no patient consent was required as the data were de-identified. The data collected from the questionnaires were manually entered into an internal database by trained research personnel. To ensure the anonymity of participants, personal identifying information, such as names, was anonymized and a unique identification number was assigned for each participant. Only authorized personnel had access to this database, and all data were stored securely and password-protected. This study was conducted in accordance with the amended Declaration of Helsinki. The study was approved by the Bioethics Committee of the Libyan Biotechnology Research Center, Tripoli, Libya, Ref No; NBC:001.H.23.10.

A COVID-19 diagnosis was made by a positive result on real time reverse transcriptase quantitative polymerase chain reaction (RT-qPCR) assay of nasopharyngeal swab specimens or positive rapid antigen test in symptomatic patients. RT-qPCR Testing was conducted at the COVID-19 diagnostic laboratories of Libyan Biotechnology Research Center, Tripoli, Libya.

Moderate to severely ill patients met at least one of the following criteria: (1) symptoms of respiratory distress with a respiratory rate 30 times/min; (2) resting blood oxygen saturation less than 94%; (3) lung imaging showing COVID-19 Reporting and Data System (CORADS) score 5. All patients admitted to the intensive care unit and medical ward were included in this study.

### Data collection and questionnaire design

Clinical data were collected from patients' records according to a questionnaire and were entered into an anonymized database. The questionnaire included personal details (age, sex, occupation, and residential address), symptoms on admission and their onset, the treatments received, whether an oxygen mask or continuous positive airway pressure (CPAP) was used, and if the patient underwent endotracheal intubation. It was also noted if the patient required renal dialysis, hemofiltration, psychiatrist intervention, developed cardiac or other complications, or if blood, plasma, or immunoglobulin transfusion was required. Information on past medical history and pre-existing illnesses, vaccination details, date of vaccination, number of doses and any side effects; previous COVID-19 infection and its severity and number of reinfections were documented. The primary outcome was in-hospital mortality. Full recovery was defined as patients who were either discharged or were still hospitalized but not ventilated at the end of the study.

## Standard treatment protocol

The treatment protocol used in this isolation center was employed in most COVID-19 care centers in Libya. ICU patients with abnormal Oxygen saturation (SpO2) and high C-reactive protein (CRP) were started on IV dexamethasone (8 mg) until SpO2 >94. Patients were treated with doxycycline tablets 200 mg daily for 10 days, vitamin C 1000mg daily, zinc tablets 60mg daily and vitamin D3 2000 IU daily. In addition, antipyretic (Paracetamol 1g) and proton pump inhibitors (pantoprazole 40 mg) were administered to most patients. Critically ill patients were given intravenous (IV) broad-spectrum antibiotics.

Supplementary high flow oxygen was administered to achieve SpO2 of ≥94% using nasal prong or face mask (simple or non-rebreathing). Patients were placed on CPAP if they failed to reach the target SpO2 with a 15L non-rebreather, and intubation with mechanical ventilation were reserved for patients whose oxygen saturations continued to decline despite non-invasive ventilation. Antiviral therapy (Remdisvir 100mg or Faviperavir 200 mg), immunosuppressants (Tocilizumab i.v/ Baricitinib 4 mg tablets), ipratropium bromide or *N*-acetylcysteine nebulizer and anticoagulants were administered to selected patients according to their clinical presentation.

## Statistical analysis

All statistical analyses were conducted using SPSS v24 software. We used descriptive statistics to characterize each cohort of patients and categorical variables were compared in relation to 5 age groups (20–35, 36–50, 51–60, 61–70 and 71–105) using Chi square analysis.

We then fitted multivariable logistic regression models with death as the dependent variable to identify factors associated with mortality. We used a backward stepwise regression model including these variables: sex, age (age was categorized as <60 years and >60 years), vaccination status and comorbidity predictors (diabetes mellitus (DM), hypertension, respiratory disease, cardiac disease, chronic kidney disease and neurological disease). Univariate analysis was also performed for individual risk factors before they were added to the multivariable model. P-values less than 0.05 were considered to be statistically significant in all analyses.

Each variable was categorized in the analysis as follows: the variable "sex" was categorized as either male or female as written in medical records. Vaccination status refers to the individuals' immunization status against COVID-19. We categorized this variable into two groups: vaccinated and unvaccinated. Participants were classified as "vaccinated" if they had received any dose of the COVID-19 vaccine, and "unvaccinated" if they had not received any doses. Comorbidity predictors are the various pre-existing medical conditions that could potentially affect the outcomes of COVID-19. We categorized these variables based on the presence or absence of each specific condition.

## Results

### Characteristics of study population

During the study period, 349 COVID-19 patients were admitted to the isolation center with. Most patients were male (211 [60.5%]) with mean± standard deviation (SD) age of 61.7 ±16.1years. Table 1 describes the clinical characteristics of the total cohort. Most patients (252 [72.2%]) had a pre-existing medical condition, including diabetes mellitus (163 patients [46.7%]), hypertension (142 patients [40.7%]), cardiac disease (44 [12.6%]), chronic kidney disease (34 patients [9.7%]), respiratory disease (23 [6.6%]), neurological disease (23 [6.6%]), liver disease (6 [1.7%]), Alzheimer's (9 [2.6%]) and cancer (5 [1.4%]). The mean±SD of number of comorbidities was 1.4± 1.3 and 19.7% of patients had >2 comorbidities.

**Table 1. General characteristics of study cohort.**

| Characteristic | Total | Age categories N (%) | | | | | P-value |
|---|---|---|---|---|---|---|---|
| | | 20–35 | 36–50 | 51–60 | 61–70 | 71–105 | |
| Mean age ± SD | | 30.15±3.95 | 44.8±4.4 | 55.8±3.04 | 65.9±2.7 | 79.8±7 | |
| All Cohorts | 349 | 20 (5.7) | 77 (22) | 66 (18.9) | 74 (21.2) | 112 (32.1) | |
| Sex | | | | | | | |
| Male | 211 | 16 (7.6) | 44 (20.8) | 47 (22.3) | 42 (20) | 62 (29.4) | 0.078 |
| Female | 138 | 4 (2.9) | 33 (23.9) | 19 (13.8) | 32 (23.2) | 50 (36.2) | |
| Previous COVID-19 diagnosis | | | | | | | |
| Yes | 4 | 2 | 0 | 0 | 2 | 0 | |
| No | 345 | 18 | 77 | 66 | 72 | 112 | |
| Comorbidity | | | | | | | |
| Yes | 252 | 7 (2.8) | 41 (16.3) | 50 (19.8) | 64 (25.4) | 90 (35.7) | <0.0001 |
| No | 97 | 13 (13.4) | 36 (37.1) | 16 (16.5) | 10 (10.3) | 22 (22.7) | |
| No of comorbidities | | | | | | | |
| 0 | 97 | 13 (13.4) | 36 (37.1) | 16 (16.5) | 10 (10.3) | 22 (22.7) | |
| 1 | 111 | 3 (2.7) | 21 (18.9) | 23 (20.7) | 29 (26.1) | 35 (31.5) | <0.0001 |
| 2 | 72 | 1 (1.4) | 8 (11.1) | 18 (25) | 18 (25) | 27 (37.5) | |
| >2 | 69 | 3 (4.3) | 12 (17.3) | 8 (11.6) | 17 (24.6) | 28 (40.5) | |
| Unvaccinated | 315 | 20 (6.3) | 74 (23.5) | 62 (19.7) | 62 (19.7) | 97 (30.8) | 0.059 |
| Vaccine dosage | | | | | | | |
| One dose | 31 | 0 | 2 (6.5) | 4 (12.9) | 12 (38.7) | 13 (41.9) | |
| Two doses | 3 | 0 | 1 (33.3) | 0 | 0 | 2 (66.7) | |
| Vaccine brand | | | | | | | |
| AstraZeneca | 9 | 0 | 1 (11.1) | 0 | 1 (11.1) | 7 (77.8) | |
| Sputnik V | 9 | 0 | 0 | 1 (11.1) | 6 (66.7) | 2 (22.2) | |
| Sinopharm | 8 | 0 | 0 | 0 | 5 (62.5) | 3 (37.5) | |
| Unknown | 8 | 0 | 2 (28.6) | 3 (42.9) | 0 | 3 (37.5) | |

The mean age of death was 64.4±15.2 years, mean age of death in patients with chronic disease was 65.7±13.7 years and the mean age of death in patients who did not have chronic disease was 59.6 ±19.1 years. The overall death rate was 57.7% (201/349), for patients 60 years or less the death rate was 75/163 (46%), and for those >60 years it was 125/185 (67.6%) [Fig 1].

Data for disease severity was only available for the prospective cohort. Slightly over half of patients were in moderate condition (51.3%), and approximately 29% of the patients in the prospective cohort were categorized as being in a severe state on admission. About 81.8% of patients with severe disease had died in comparison with 20% and 33.3% for those with mild and moderate disease respectively.

## Comparison of vaccinated and unvaccinated COVID-19 patients

The vast majority of patients in our study were not vaccinated against COVID-19 (315 [90.3%]) and were not previously infected (345/349 [98.9%]) (Table 2). The proportion of individuals who received one dose of the vaccine was 14.1% (30/213), while the proportion of individuals who received two doses of the vaccine was 1.4% (3/213). A comparison of vaccination rate between genders revealed a statistically significant difference, with 13.3% (28/211) of male patients being vaccinated in contrast to only 4.3% (6/138) of female patients (p = 0.006).

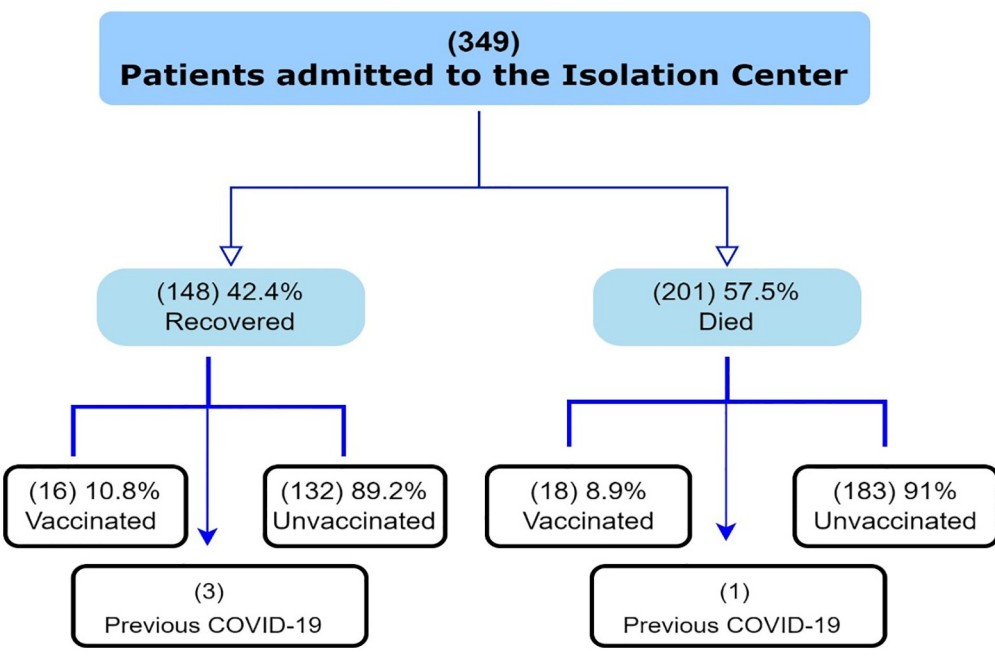

**Fig 1. Outcome of study cohort.**

The 3 patients who received two doses of vaccine recovered and were discharged from the hospital. Two patients got Sinopharm and AstraZeneca vaccines, while the third patient received an unknown type of vaccine. Two had pre-existing medical conditions. The three individuals were aged between 47 and 78 years old.

**Table 2. Comparison of hospitalized patients during study period (May–October 2021) by vaccination status.**

| Characteristic | Vaccinated (%) | Unvaccinated (%) | P-value |
|---|---|---|---|
| N | 34 | 315 | |
| Sex (Male) | 28 (82.4) | 183 (58.1) | 0.006 |
| Age, years | 68.2±12.5 | 61±16.3 | 0.013 |
| Pre-existing medical conditions | 27 (79.4) | 225 (71.4) | 0.324 |
| No of comorbidities | | | |
| 0 | 15 (44.1) | 90 (28.6) | |
| 1 | 11 (32.4) | 98 (31.1) | |
| 2 | 4 (11.8) | 65 (20.6) | |
| >2 | 4 (11.8) | 62 (19.7) | |
| Hypertension | 16 (47.1) | 126 (40) | 0.426 |
| Diabetes mellitus | 19 (55.9) | 144 (45.7) | 0.259 |
| Cardiac disease | 7 (20.6) | 37 (11.7) | 0.140 |
| Chronic kidney disease | 3 (8.8) | 31 (9) | 0.849 |
| Respiratory disease | 1 (2.9) | 22 (7) | 0.367 |
| Neurological disease | 1 (2.9) | 22 (7) | 0.367 |
| Previous COVID-19 | 1 (2.9) | 3 (1) | 0.301 |
| Death | 18 (52.9) | 183 (58.1) | 0.56 |
| Number of doses: | | | - |
| One vaccine dose | 31 (91.2) | N/A | |
| Two vaccine doses | 3 (8.8) | N/A | |

The mortality rate among patients who received only one dose of any COVID-19 vaccine was 58.1% (18/31). Of those patients, a majority of the individuals who died were male (77.8%, 14/18) and had an age above 60 years (89%, 16/18). Furthermore, 77.8% (14/18) of the deceased had pre-existing medical conditions. Among these patients with comorbidities, 50% (9/18) had hypertension, 55.6% (10/18) had diabetes, 16.7% (3/18) had cardiac disease, and 5.5% (1/18) had chronic kidney disease. The remaining four individuals (4/18) did not have any comorbidities. Two of them received the AstraZeneca vaccine, one had Sputnik V and the fourth had an unknown type of vaccine. These patients were aged between 58–85 years old. One of them received the vaccine just three days before hospital admission, while the other three received it 78–116 days before admission.

On the other hand, no mortality was found among those who received two doses of any vaccine. Among unvaccinated patients the mortality was 183/315 (58.1%).

Of the patients who received one dose of the vaccine, one had a previous COVID-19 infection. This individual was a male >65 years and died due to cardiac complications. His previous COVID-19 infection was mild, characterized by loss of smell and taste. The patient received one dose of the Sputnik V vaccine approximately 57 days prior to his illness.

Among patients vaccinated with one dose of vaccine who had medical comorbidities and died (14), 4 were vaccinated with AstraZeneca, 4 with Sputnik V, and 4 with Sinopharm. Two patient didn't report the vaccine type. The duration between vaccination and admission was less than 45 days in 7 patients and >45 days in 5 patients.

Among those vaccinated with one dose who had full recovery (13/31), the majority were men (11/13) with a mean age of 64.6 years. Four had received Sputnik V, 2 patients had Astra-Zeneca, 3 had Sinopharm and 4 had unknown vaccine type. About 85% (11/13) had at least one comorbidity including hypertension, diabetes, chronic kidney disease and cardiac disease.

Also, among patients with a prior COVID-19, 75% (3/4) had complete recovery and 25% (1/4) died from COVID-19 during their hospital stay.

## Comparison of different periods of the COVID-19 pandemic

The mortality rates were calculated for each month as follows: 33.3% in May, 48.8% in June, 63.1% in July, 78.3% in August and 44.7% in September and October (p value≤0.0001) [Fig 2]. From mid-June to October, infections in Libya were caused primarily by the Delta strain [16].

## Predictors of mortality

In univariate analysis, the factors most strongly associated with death were age >60 years (OR = 2.46, 95% confidence interval 1.6–3.8, p-value = <0.0001), female sex (OR = 1.78, 95% CI 1.14–2.7, p-value = 0.011) and hypertension (OR = 0.6, 95% CI 0.39–0.94, p-value = 0.025). In addition, univariate analysis showed that mortality increases with increasing number of pre-existing medical conditions. The odds of death increases 2.8 fold in patients with >2 medical conditions (p-value 0.008) compared with those who do not have a chronic disease on presentation. The odds of death increased 2 fold in those with at least 1 comorbid condition (p-value = 0.010).

We constructed a multivariable prediction model using sex, age, chronic diseases, and vaccination status as possible risk factors for the patient cohort Table 3. The model showed that the only significant factor was age older than 60 years (OR = 2.328, CI 1.456–3.724, p-value = <0.0001).

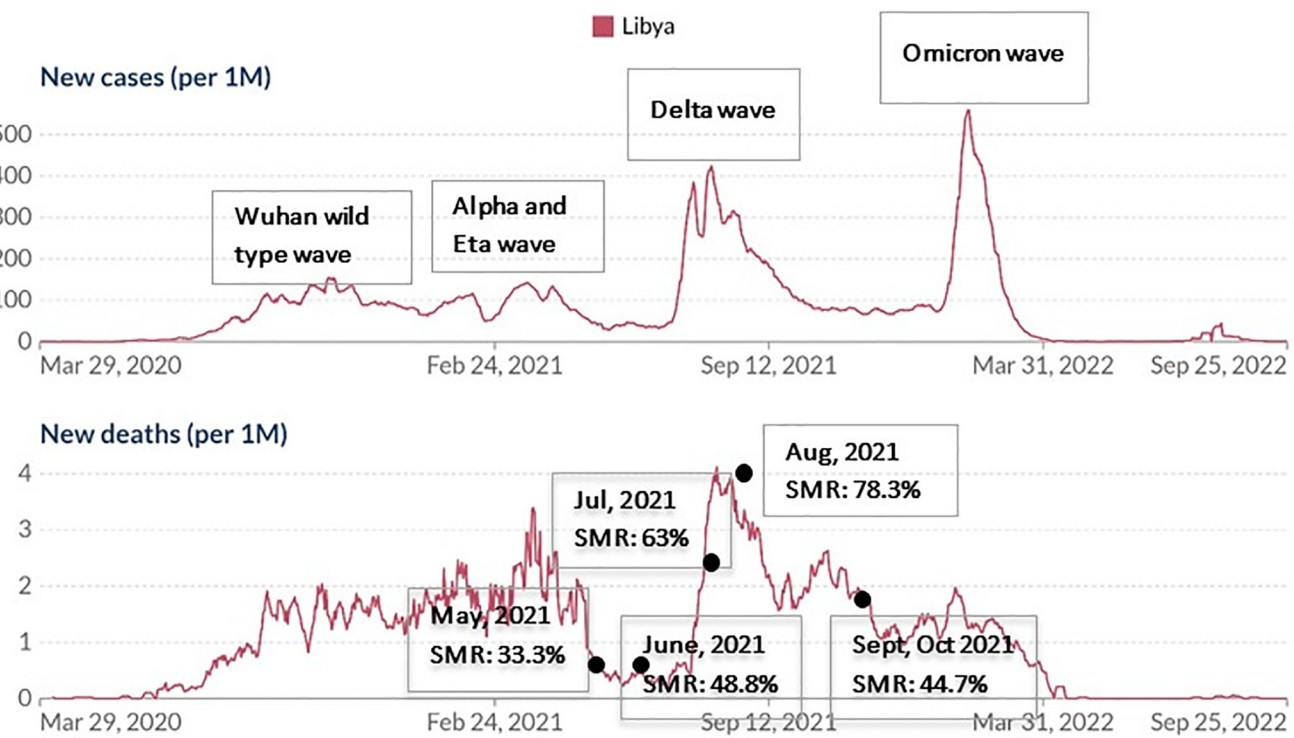

**Fig 2. Seven-day rolling average of COVID-19 new confirmed cases in Libya and mortality rates in 2021.** The figure shows the different COVID-19 epidemic peaks in Libya and hospital mortality rates in this study. SMR = study mortality rate.

## Discussion

This study describes the outcome and clinical characteristics of 349 patients with COVID-19 admitted to Souq Thullatha Isolation Center, a COVID-19 dedicated care center in Tripoli, Libya during the Delta wave.

Our study showed that 57.5% of patients died due to complications of COVID-19 during their hospital stay. The mortality rate reported in other studies ranged from 8.7% to 54.6% among those patients with COVID-19 pneumonia [11, 17–19]. The high mortality rate and poor ICU outcomes at some medical centers led to concerns regarding the effectiveness of standard of care delivery and mechanical ventilation measures [20].

**Table 3. Multivariable risk model for mortality.**

| Characteristic | Adjusted Odds ratio (95% CI) | *P*-value |
|---|---|---|
| Sex (Female) | 1.550 (0.970–2.477) | 0.067 |
| Age >60 | 2.328 (1.456–3.724) | <0.0001 |
| Hypertension | 0.789 (0.478–1.302) | 0.354 |
| Diabetes Mellitus | 0.837 (0.528–1.326) | 0.448 |
| Chronic kidney disease | 0.991 (0.454–2.166) | 0.983 |
| Cardiac disease | 1.123 (0.551–2.288) | 0.750 |
| Respiratory disease | 0.655 (0.250–1.716) | 0.390 |
| Neurological disease | 0.787 (0.303–2.041) | 0.622 |
| Vaccination status | 1.450 (0.677–3.106) | 0.339 |

Most patients admitted at our center were either not vaccinated (90%) nor previously infected with SARS-COV-2 (98.8%). This suggests that previous infection (naturally induced immunity) and full vaccination against COVID-19 significantly reduces hospitalization and death. Furthermore, there was no mortality among fully vaccinated patients. Our cohort only had 3 fully vaccinated patients which suggests that two doses of vaccine probably protected individuals from severe disease warranting admission. Furthermore, no mortality was seen among those vaccinated with two vaccine doses. At the time of conducting this study, the Libyan COVID-19 vaccination program had already started in April 2021 and by October 2021 about 1,393,030 persons had received one dose of vaccine and 211,278 had received two doses according to the NCDC data. The total number of infected individuals in the NCDC registry was 341,091 on September 30[th] 2021. Our results confirm the findings of previous studies on the protective effects of full vaccination, whereas immunity from a single dose of vaccine tends to wane faster and may not mount a good immune response after infection except in previously infected individuals [4]. We also found no difference in mortality between unvaccinated and partially vaccinated patients. This is in agreement with other studies that showed a significant difference in mortality between fully vaccinated and either unvaccinated or partially vaccinated individuals [21]. Our results indicate that if a person has a history of infection or is only partially vaccinated, if they are admitted with severe disease, the risk of death is essentially the same as unvaccinated naïve individuals. We recommend that treatment should be sought at the stage of mild or moderate disease, since progression to severe disease leads to adverse outcomes.

In addition, the mortality rates in this study were highest in July and August coinciding with the delta predominance in most countries [22–24]. Mortality rates then moderately decreased in September and October. This is probably due to increased vaccination coverage and provision of a second dose to people who had Sputnik V as their first dose. mRNA COVID-19 vaccination had also started in early August enhancing the vaccine resources in Libya [25].

In our study cohort, 72.2% of patients had at least one pre-existing condition and about 20% had >2 comorbidities. Other studies showed similar findings, the proportion of patients with at least one of eight chronic diseases ranged between 65–80% of patients admitted to hospitals [26, 27]. Our results showed that a high percentage of patients were of advanced age with comorbidities, diabetes and hypertension being the most common. They were also more likely to be male. These findings support the observations of other studies [10, 11, 28].

Studies showed that black or African American race, male sex, severe obesity and chronic kidney disease were significantly associated with the need for ICU care [10, 27, 29]. Furthermore, previous studies showed that patients admitted with pre-existing disease especially heart failure and chronic kidney disease, old age and male sex have worse prognosis [26, 30]. However, in our study there was no statistically significant difference between male and female sex in the probability of death when sex was adjusted with other factors (Table 2). When univariate analysis was carried out for each risk factor female patients were more likely to die than male patients. These differences could be explained by lower vaccination coverage in our female cohort (4.3% vs 13.3%). Individual traits including age, obesity, and pre-existing illness, are known to have an impact on immunological competence [31].

The vaccination campaign in Libya started on April 17[th], 2021 with the first doses of the AstraZeneca and Sputnik V vaccines being given. On August 26[th], 2021, 1,035,640 persons had received the first vaccine dose, while 32,693 had received two doses. Unfortunately, many people who received the first dose of Sputnik V didn't receive the second dose until October-December and most of them received Astrazeneca as a second dose. In addition, Pfizer-BioNtech vaccination started later in August.

Our study period (mid-June to October) coincided with the emergence of the Delta SARS--COV-2 variant [16]. SARS-COV-2 samples from Libya were regrettably not confirmed by genome sequencing during that time; instead, tracking relied on RT-qPCR assays that targeted the presence of the L452R and P681R mutations. However, genome sequencing was performed on samples collected around the end of February and beginning of March 2021, and the majority of cases were B.1.525, with only 2% being an alpha variant [32]. Hence, we believe that most of SARS-COV-2 infection in May and beginning of June 2021 was B.1.525 variant.

It is important to note that the political and economic instability in Libya has hugely affected the response to the pandemic, especially the provision of timely vaccine doses, oxygen supplies, capacity building for physicians, and the availability of antiviral and supportive therapies.

The risk models created in this study may be used in a variety of health care settings during the pandemic to assess individual risk for death and guide patient management.

Health recording systems in isolation centers should be improved and all patient clinical details should be registered. This could be achieved by strong vigilance, auditing, and establishment of an electronic health record system. Regular reporting to relevant authorities is suggested to quickly identify missing data files. High mortality rates raised concerns on the effectiveness of standard of care delivery. Health sector improvement plans should strongly consider physicians' capacity building in critical care and pandemic preparedness among the priority reform strategies.

Our study has a number of limitations. First, this study was an observational study conducted at a single center in the Tripoli area, thus the results might not reflect the situation in other Libyan COVID-19 isolation centers. Also, some of patient medical records were missing the brand of COVID-19 vaccine.

We also recognize that the sample size may have some limitations. However, in our study, we made every effort to include as many participants as possible within the available resources and timeframe. The combined retrospective and prospective cohorts provided a substantial number of cases, allowing us to analyze the impact of risk factors and vaccination on mortality from COVID-19. However, while the inclusion of all patients in the isolation center allowed for a comprehensive analysis of the impact of vaccination on mortality, the retrospective cohort lacks specific data on the exact number of severe and moderate cases. Although precise proportions were not available in the retrospective cohort, the inclusion of all patients from the isolation center allowed us to capture a representative sample of severe and moderate cases. In Libya, most COVID-19 patients initially visit isolation centers as outpatients to receive treatment. However, many patients refuse to be admitted, preferring to continue their care at home. This is often because they do not want to be isolated from their families. As a result, many patients who are admitted to the centers are already moderately to severely ill, having delayed seeking medical care.

Additionally, although a multivariate regression analysis was performed to study multiple factors associated with mortality, potential confounders may still be unrecognized. Our information regarding patient death is limited to hospital stay and patients were not followed-up after discharge. Nevertheless, our results provide important data on mortality rates and associated risk factors and help inform public health COVID-19 control efforts. Further investigations should study other unmeasured factors including inflammatory markers to predict factors associated with mortality in COVID-19 patients admitted to the ICU.

## Conclusion

Previous infection or full vaccination against COVID-19 significantly reduces rates of hospitalization and death, as most admitted patients were unvaccinated and not previously infected.

However, a single vaccine dose may not be adequate, especially for older individuals and those with underlying medical conditions.

Overall, we found that age and medical comorbidities were strong predictors of mortality. Clinicians should consider older patients with comorbidities as very high risk and treat them with maximal care. High-risk older patients with comorbidities should be fully vaccinated and offered up to date bivalent COVID-19 booster doses.

## Supporting information

**S1 File.**
(XLSX)

## Acknowledgments

The authors would also like to thank Dr. Hannah Gendelman and Dr. Eric Guardino (Harvard University) for their English language revision. While it is not possible to honor all of the Libyan health workers who have died from COVID-19, the authors would like to pay special tribute to Dr. Firas Fadel who lost his life to the disease while serving his patients. Their contributions have helped to improve our paper and made it accessible to a wider audience. Finally, the authors would like to thank the Libyan Ministry of Health and the Libyan Biotechnology Research Center for their support.

## Author Contributions

**Conceptualization:** Inas Alhudiri, Zakarya Abusrewil, Mohammed Ammar Awn.

**Data curation:** Inas Alhudiri, Zakarya Abusrewil, Omran Dakhil, Mosab Ali Zwaik, Mohammed Ammar Awn, Aimen Ibrahim Ahmed.

**Formal analysis:** Inas Alhudiri, Rasha Abugrara.

**Methodology:** Inas Alhudiri, Zakarya Abusrewil, Aimen Ibrahim Ahmed.

**Project administration:** Inas Alhudiri, Zakarya Abusrewil, Mohammed Ammar Awn.

**Resources:** Zakarya Abusrewil, Omran Dakhil, Mosab Ali Zwaik.

**Supervision:** Inas Alhudiri, Zakarya Abusrewil, Adam Elzagheid.

**Writing – original draft:** Inas Alhudiri, Zakarya Abusrewil, Mwada Jallul.

**Writing – review & editing:** Inas Alhudiri, Zakarya Abusrewil, Omran Dakhil, Mosab Ali Zwaik, Mohammed Ammar Awn, Adam Elzagheid.

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
