## [Decision Letter · Decision Letter 0]

24 Apr 2023

PONE-D-23-02510Impact of vaccination and risk factors on COVID-19 mortality amid delta wave in Libya: a single centre cohort studyPLOS ONE

Dear Dr. Alhudiri,

Thank you for submitting your manuscript to PLOS ONE. After careful consideration, we feel that it has merit but does not fully meet PLOS ONE’s publication criteria as it currently stands. Therefore, we invite you to submit a revised version of the manuscript that addresses the points raised during the review process. The comments of editor and reviewers are submitted below

We look forward to receiving your revised manuscript.

Kind regards,

Marwa Shawky Abdou, DPH

Academic Editor

PLOS ONE

Journal Requirements:

 Whilst you may use any professional scientific editing service of your choice, PLOS has partnered with both American Journal Experts (AJE) and Editage to provide discounted services to PLOS authors. Both organizations have experience helping authors meet PLOS guidelines and can provide language editing, translation, manuscript formatting, and figure formatting to ensure your manuscript meets our submission guidelines. To take advantage of our partnership with AJE, visit the AJE website (http://aje.com/go/plos) for a 15% discount off AJE services. To take advantage of our partnership with Editage, visit the Editage website (www.editage.com) and enter referral code PLOSEDIT for a 15% discount off Editage services. If the PLOS editorial team finds any language issues in text that either AJE or Editage has edited, the service provider will re-edit the text for free.

Additional Editor Comments:

- Abstract better to be structured

- English language need to be revised

- The objective need to be more clear

- The sample size is not clear how had been calculated

- (Moderate to severe patients who meet at least one of the following criteria: (1) symptoms of respiratory distress with a respiratory rate 30 times/min; (2) resting blood oxygen saturation less than 94%; (3) the imaging examination of lungs showing CORADS5 score). The previous sentence is not clear why it is added in the manuscript? Are they inclusion/exclusion criteria?

- The standard treatment protocol, Is it the treatment protocol in the treatment center for COVID or the protocol in the study? If so, how treatment protocol in cohort study?

- The result section should be constructed according to the journal guidelines

- The mean age of participants was written by two different ways: mean (SD) and mean ± SD, please use one way of them

- What is the SD of age of death?

- What are the limitations of the study?

- References needed to be revised according to journal guidelines

Reviewers' comments:

Reviewer's Responses to Questions

**Comments to the Author**

1. Is the manuscript technically sound, and do the data support the conclusions?

Reviewer #1: Yes

Reviewer #2: Partly

2. Has the statistical analysis been performed appropriately and rigorously? 

Reviewer #1: Yes

Reviewer #2: Yes

3. Have the authors made all data underlying the findings in their manuscript fully available?

Reviewer #1: Yes

Reviewer #2: Yes

4. Is the manuscript presented in an intelligible fashion and written in standard English?

Reviewer #1: No

Reviewer #2: Yes

5. Review Comments to the Author

Reviewer #1: The manuscript needs to be copy-edited by a native English speaker.

Please ensure consistency in how percentages are reported, keep either up to one decimal place or round off to the nearest whole number. Currently, the manuscript has a mix of both.

The authors include the criteria for identifying moderate to severe cases of COVID-19 in the methods section but then say that they included all patients in the isolation center. How is the criteria relevant here?

Please make sure that all abbreviations are introduced in full form the first time they are mentioned.

Under "Data collection and questionnaire design" - please specify the 'personal details' that were collected and the database into which the data was entered and how the names were anonymized.

Clarify whether forward or backward stepwise regression was used in the methods section. Also clarify the rationale for having two regression models - one for all ages and one for 60 years or under.

Clarify how each variable is categorized in your analysis- sex, vaccination status and comorbidity predictors in the methods section.

In table 1, please include subheadings under 'any vaccine' -the first two variables are 'dosage' and the ones that follow are 'brands of vaccines'.

In the results section, please report mortality rates among those vaccinated with a single dose, two doses and those unvaccinated and organize the paragraphs accordingly. Currently, only that of those vaccinated with a single dose is reported.

After the following sentence in the discussion section "Our results indicate that whether a person has a history of infection or is only partially vaccinated, once admitted with severe disease, the risk of death is essentially the same as unvaccinated naïve individuals" include a recommendation like treatment should be sought at the stage of mild or moderate disease, since progression to severe disease leads to adverse outcomes.

"Studies showed that race, male sex, severe obesity and chronic kidney disease were significantly associated with the need for ICU care" -specify race.

Reviewer #2: it is a very well written article. I have shared some questions that need to be addressed before accepting

- What was the reason for selecting a prospective and a retrospective cohort? Please explain the difference of doing both

- How the participants were selected to be part of cohort

- How was the sample size derived?

- Do you think the sample is enough for a cohort study – this might me a limitation?

- Was any randomization done at any stage during sample selection?

- What was the process of taking consent from the participants in accessing the medical records

6. PLOS authors have the option to publish the peer review history of their article (what does this mean?). If published, this will include your full peer review and any attached files.

Reviewer #1: **Yes: **Lamisa Ashraf

Reviewer #2: **Yes: **Mariam Ashraf

---

## [Author Response · Author response to Decision Letter 0]

8 Jun 2023

Dear Editor and reviewers

Thank you very much for your time and effort in reviewing our manuscript. The authors would like to thank the editor for their careful handling of the manuscript and for their helpful suggestions and the reviewers for their thoughtful comments and for their insights into our work. We appreciate your thoughtful comments and suggestions, which have been very helpful in improving the quality of our paper. We have carefully considered all of your comments and made the required revisions. 

Please find our responses written in bold green and color below each reviewer comment. If any responses are unclear or you wish additional changes, please let us know.

Sincerely,

Inas 

Response to the editor’s comments:

1. Abstract better to be structured

The abstract is structured as requested into introduction, methods, results and conclusion.

2. English language need to be revised

Two native English speakers (Clinician scientists) kindly revised our manuscript (Dr. Eric Guardino and Dr. Hannah Gendelman). The comments and suggestions of both were used. Their names were included in the acknowledgment and in the submission system.

3. The objective need to be more clear

The objectives of this study were rewritten in a more clear statement at the end of the introduction.

4. The sample size is not clear how had been calculated

We recruited the entire accessible adult population (patients) admitted to the isolation center in ICU and ward during the study period (May-October 2021), therefore no sample size calculations were required. In addition, given the specific circumstances during the delta wave of COVID-19, it was aimed to gather data on the admitted patients during that period. 

5. (Moderate to severe patients who meet at least one of the following criteria: (1) symptoms of respiratory distress with a respiratory rate 30 times/min; (2) resting blood oxygen saturation less than 94%; (3) the imaging examination of lungs showing CORADS5 score). The previous sentence is not clear why it is added in the manuscript? Are they inclusion/exclusion criteria?

We wanted to define the terms "moderate to severe" as used by the isolation center in the study. However, these terms were not inclusion or exclusion criteria, as all admitted adult patients were enrolled in the study. Unfortunately, data on severity at admission was only available for the prospective cohort. The number of patients with severe, moderate, and mild disease for the prospective cohort has now been added to the results section, along with the proportions of deaths in each category.

6. The standard treatment protocol, Is it the treatment protocol in the treatment center for COVID or the protocol in the study? If so, how treatment protocol in cohort study?

The standard treatment protocol written in methods section was the protocol used in the isolation center of our study. This protocol was based on the best available evidence and consistent with the standard treatment protocol for COVID-19 used in most centres in Libya. The reason for adding the details about the standard treatment protocol in our manuscript was to ensure that all patients in the study receive the same level of care. This helps to minimize bias in the study results and ensures that the findings are generalizable to other patients with COVID-19.

7. The result section should be constructed according to the journal guidelines

Done as suggested. 

8. The mean age of participants was written by two different ways: mean (SD) and mean ± SD, please use one way of them

Thank you for your note. It was corrected accordingly. Mean ± SD was used throughout the paper.

9. What is the SD of age of death?

The SD of age of death was added in the results.

(The overall mean age of death was 64.4±15.2 years, mean age of death in patients with chronic disease was 65.7±13.7 years and the mean age of death in patients who did not have chronic disease was 59.6 years±19.1.)

10. What are the limitations of the study?

The limitations of the study was written at the end of the discussion as a separate paragraph.

11. References needed to be revised according to journal guidelines 

References were revised and formatted according to the journal guidelines

Reviewer #1: 

The manuscript needs to be copy-edited by a native English speaker.

Author

Two native English speakers had kindly revised the manuscript (Dr. Eric Guardino and Dr. Hannah Gendelman)

1. Please ensure consistency in how percentages are reported, keep either up to one decimal place or round off to the nearest whole number. Currently, the manuscript has a mix of both.

Author 

Thank you for your comment. The percentages were unified. 

2. The authors include the criteria for identifying moderate to severe cases of COVID-19 in the methods section but then say that they included all patients in the isolation center. How is the criteria relevant here?

Author

Thank you for your feedback. The inclusion of the criteria for identifying moderate to severe cases in the methods section was intended to provide transparency regarding our methodology. By describing the criteria, we aimed to ensure that readers have a clear understanding of how we classified the severity of COVID-19 cases. 

 In our study, we acknowledge that in both the retrospective and prospective cohorts, a substantial proportion of patients were classified as either severe or moderate cases of COVID-19. This observation is consistent with the situation in our country, where patients often seek medical help at a later stage of COVID-19 disease. We have also added these data for prospective cohort in the manuscript in the results section. Although precise proportions were not available in the retrospective cohort, the inclusion of all patients from the isolation center allowed us to capture a representative sample of severe and moderate cases. Most patients in Libya come to the COVID-19 isolation centers as outpatient to receive appropriate treatment and then continue management at home, because the refuse admission and they don’t want to be isolated and kept away from their families. For this reason, most admitted patients are moderately to severely diseased.

We have also updated the limitations paragraph at the end of the discussion in the manuscript and stated that while the inclusion of all patients in the isolation center allowed for a comprehensive analysis of the impact of vaccination on mortality, the retrospective cohort lacks specific data on the exact number of severe and moderate cases.

3. Please make sure that all abbreviations are introduced in full form the first time they are mentioned.

Author

Thank you. We have made the necessary revisions to ensure that all abbreviations are introduced in their full form the first time they are mentioned.

4. Under "Data collection and questionnaire design" - please specify the 'personal details' that were collected and the database into which the data was entered and how the names were anonymized. 

Author

Thank you for your comment. We collected various personal details from the participants. These details included demographic information such as age, sex, occupation, and residential address. Additionally, we collected medical history information, including any pre-existing conditions or comorbidities.

Regarding the database used for data entry, we did not use a public database for this study. Instead, we employed a secure, internal database system. The data collected from the questionnaires were manually entered into this internal database by trained research personnel. To ensure the anonymity of participants, personal identifying information, such as names, was anonymized and a unique identification number was assigned for each participant. Only authorized personnel had access to this database, and all data were stored securely and password-protected. All these details were updated in the manuscript.

5. Clarify whether forward or backward stepwise regression was used in the methods section. 

Author

Backward stepwise regression was used for logistic regression analysis. It was clarified in the methods section.

 Also clarify the rationale for having two regression models - one for all ages and one for 60 years or under. 

Author

That mention of the construction of two models was a mistake and was not intended to be included in the final submitted version of the paper. We regret any confusion it may have caused and assure you that it has been removed from this final version and only the results of one model (for all ages) are included in the manuscript. However, regarding the rationale for constructing two regression models, we initially considered examining the impact of different risk factors on mortality separately for different age groups to assess any potential age-specific effects. In the first model, we evaluated the association between age and mortality for all individuals and for the age variable it was categorized into >60 and < 60 years old and this analysis revealed a significant impact of age on mortality. However, in subsequent analyses of other variables within both the overall cohort and the young group cohort (<60 years of age), no significant effects on mortality were observed. 

Based on these findings, we concluded that the inclusion of the results from the model for individuals aged 60 years or under was not necessary for the final version of the manuscript, as it did not provide additional insights beyond the significant association found in the initial model for all individuals.

6. Clarify how each variable is categorized in your analysis- sex, vaccination status and comorbidity predictors in the methods section.

Author

Thank you for your comment. In the methods section of our manuscript, we have now described the categorization of each variable used in our analysis based on your suggestion. Each variable was categorized as follows:

Sex: The variable "sex" represents the biological sex of the participants and is categorized as either male or female. We collected this information from medical records. 

Vaccination status: The variable "vaccination status" refers to the individuals' immunization status against COVID-19. We categorized this variable into two groups: vaccinated and unvaccinated. Participants were classified as "vaccinated" if they had received any dose of the COVID-19 vaccine, and "unvaccinated" if they had not received any doses. When only one dose or two doses were received, it was clearly mentioned. Comorbidity predictors: Comorbidity predictors are the various pre-existing medical conditions that could potentially affect the outcomes of COVID-19. We categorized these variables based on the presence or absence of each specific condition. For example, if a participant had diabetes, they would be categorized as having "diabetes" (present) or "no diabetes" (absent).

7. In table 1, please include subheadings under 'any vaccine' -the first two variables are 'dosage' and the ones that follow are 'brands of vaccines'.

Author

 Thank you for this comment. Done as suggested.

8. In the results section, please report mortality rates among those vaccinated with a single dose, two doses and those unvaccinated and organize the paragraphs accordingly. Currently, only that of those vaccinated with a single dose is reported.

Author

Thank you for your suggestion. There was no mortality among those who received two doses of the vaccine. We have organized the paragraphs as requested and added the mortality rates for all the above categories. 

9. After the following sentence in the discussion section "Our results indicate that whether a person has a history of infection or is only partially vaccinated, once admitted with severe disease, the risk of death is essentially the same as unvaccinated naïve individuals" include a recommendation like treatment should be sought at the stage of mild or moderate disease, since progression to severe disease leads to adverse outcomes.

Author

Thank you for your suggestion to include a recommendation in the discussion section following that sentence. According your suggestion, the recommendation was incorporated into the discussion section.

10. "Studies showed that race, male sex, severe obesity and chronic kidney disease were significantly associated with the need for ICU care" -specify race. 

Author

The race was specified in the manuscript in the same sentence according to how the authors of the cited studies had specified it. Another reference was added which reported that black and African American race was significantly associated with the need for ICU admission (Auld SC, Harrington KRV, Adelman MW, Robichaux CJ, Overton EC, Caridi-Scheible M, et al. Trends in ICU Mortality From Coronavirus Disease 2019: A Tale of Three Surges. Crit Care Med. 2022;50: 245–255. doi:10.1097/CCM.0000000000005185)

Reviewer #2:

 It is a very well written article. I have shared some questions that need to be addressed before accepting

1. What was the reason for selecting a prospective and a retrospective cohort? Please explain the difference of doing both.

Author

Thank you for your comment. The rationale behind incorporating both retrospective and prospective cohorts was to enhance the robustness of our findings and increase the overall sample size. The retrospective cohort was from May to August 2021, while the prospective cohort covered the period from end August to October 2021. This combination allowed us to obtain a larger number of cases and overcome potential limitations associated with studying a single cohort and increase the generalizability of our findings. 

The inclusion of the retrospective cohort provided valuable historical data covering pre- Delta and Delta wave, while the prospective cohort allowed us to gather more information on patient outcomes at the peak of Delta and beyond. In addition, we consulted a statistician and epidemiologist and they said it is appropriate to combine both cohorts as long as they are independent and exposed to the same factors. We added a detailed explanation in the methods section of the manuscript to clarify the rationale behind combining the retrospective and prospective cohorts. 

2. How the participants were selected to be part of cohort

Author

Thank you for your question. Since we wanted to study the impact of vaccination on COVID-19, we have included all adult patients admitted to the isolation center. So inclusion criteria was adults (>18 years old) and SARS-COV-2 positive patients admitted to the isolation center. Subsequently no randomization was performed.

3. How was the sample size derived?

Author

We recruited the entire accessible adult population (patients) admitted to the isolation center in ICU and ward during the study period (May-October 2021), therefore no sample size calculations were required. In addition, given the specific circumstances during the delta wave of COVID-19, it was aimed to gather data on the admitted patients during that period.

4. Do you think the sample is enough for a cohort study – this might me a limitation?

Author

We recognize that the sample size may have some limitations. However, in our study, we made every effort to include as many participants as possible within the available resources and timeframe. The combined retrospective and prospective cohorts provided a substantial number of cases, allowing us to analyze the impact of risk factors and vaccination on mortality from COVID-19.

5. Was any randomization done at any stage during sample selection?

Author

No randomization was performed as I mentioned above because we enrolled all the patients admitted to the center.

6. What was the process of taking consent from the participants in accessing the medical records?

Author

The retrospective and prospective design of our study relied on de-identified and anonymized data from medical records, removing any personal identifiers that could potentially link the data back to individual patients and ensuring the confidentiality and privacy of the patients involved, however consent was not needed for the retrospective cohort.

For the prospective component of our study, we followed strict ethical guidelines and procedures to obtain informed consent from the participants. In cases where participants were unable to provide informed consent due to their severe medical condition, we sought consent from their legal representatives or next of kin.

A clear explanation in the methods section of our manuscript was written. Before enrolling any individuals in the study, we provided detailed information about the study's objectives, procedures, potential risks, and benefits. To adhere to ethical guidelines and regulations, we obtained approval from the appropriate ethics committee prior to conducting the study. The ethics committee reviewed the study protocol, including the data collection procedures and the use of medical records, and approved the study with the understanding that patient privacy and confidentiality would be maintained throughout the research process.

---

## [Decision Letter · Decision Letter 1]

20 Jul 2023

Impact of vaccination and risk factors on COVID-19 mortality amid delta wave in Libya: a single center cohort study

PONE-D-23-02510R1

Dear Dr. Alhudiri,

We’re pleased to inform you that your manuscript has been judged scientifically suitable for publication and will be formally accepted for publication once it meets all outstanding technical requirements.

Kind regards,

Marwa Shawky Abdou, DPH

Academic Editor

PLOS ONE

Additional Editor Comments (optional):

Reviewers' comments:

Reviewer's Responses to Questions

**Comments to the Author**

1. If the authors have adequately addressed your comments raised in a previous round of review and you feel that this manuscript is now acceptable for publication, you may indicate that here to bypass the “Comments to the Author” section, enter your conflict of interest statement in the “Confidential to Editor” section, and submit your "Accept" recommendation.

Reviewer #1: All comments have been addressed

Reviewer #2: All comments have been addressed

2. Is the manuscript technically sound, and do the data support the conclusions?

Reviewer #1: Yes

Reviewer #2: Yes

3. Has the statistical analysis been performed appropriately and rigorously? 

Reviewer #1: Yes

Reviewer #2: Yes

4. Have the authors made all data underlying the findings in their manuscript fully available?

Reviewer #1: Yes

Reviewer #2: No

5. Is the manuscript presented in an intelligible fashion and written in standard English?

Reviewer #1: Yes

Reviewer #2: Yes

6. Review Comments to the Author

Reviewer #1: (No Response)

Reviewer #2: (No Response)

7. PLOS authors have the option to publish the peer review history of their article (what does this mean?). If published, this will include your full peer review and any attached files.

Reviewer #1: **Yes: **Lamisa Ashraf

Reviewer #2: **Yes: **Mariam Ashraf

---

## [Editor Report · Acceptance letter]

27 Jul 2023

PONE-D-23-02510R1 

Impact of vaccination and risk factors on COVID-19 mortality amid delta wave in Libya: a single center cohort study 

Dear Dr. Alhudiri:

I'm pleased to inform you that your manuscript has been deemed suitable for publication in PLOS ONE. Congratulations! Your manuscript is now with our production department. 

Kind regards, 

on behalf of

Dr. Marwa Shawky Abdou 

Academic Editor

PLOS ONE